# The Role of Alcohols in the Hexene-1 Hydroalkoxycarbonylation Reaction with Catalysts Based on Palladium Complexes

Gulbanu Zhaksylykova [1], Kairzhan Shalmagambetov [1], Fatima Kanapiyeva [1], Nurbolat Kudaibergenov [1,*], Marat Bulybayev [1], Meruyert Zykai [1], Gulmira Abyzbekova [2] and Gulzhan Balykbayeva [2]

[1] Center of Physical Chemical Methods of Research and Analysis, Faculty of Chemistry and Chemical Technology, Al-Farabi Kazakh National University, Al-Farabi Ave. 71, Almaty 050038, Kazakhstan; banu.81@mail.ru (G.Z.); kairshan@yandex.ru (K.S.); fatima31@mail.ru (F.K.); marat.bulybayev14@gmail.com (M.B.); mzykay@mail.ru (M.Z.)

[2] Department of Biology, Geography and Chemistry, Korkyt Ata Kyzylorda University, Aiteke Bi Str. 29A, Kyzylorda 120014, Kazakhstan; abizgul@mail.ru (G.A.); bal_gulzhan@mail.ru (G.B.)

[*] Correspondence: nurbolat.kudaibergenov@kaznu.kz or n.zh.kudaibergenov@gmail.com

**Abstract:** In this work, the activity of various alcohols in the hexene-1 hydroalkoxycarbonylation reaction in the presence of two catalytic systems was investigated for the first time: (1) $Pd(PPh_3)_4$-$PPh_3$-TsOH (menthol, cyclohexanol, ethanol, propanol, iso-propanol, butanol, isobutanol and benzyl alcohol) and (2) $PdCl_2(PPh_3)_2$-$PPh_3$-$AlCl_3$ (ethanol, propanol-1, butanol-1, isoamyl alcohol, isobutanol, pentanol-1, allyl alcohol and tert-butyl alcohol). The optimal process parameters (temperature, pressure and reaction time) for the reactions of the hydropropoxycarbonylation and hydrobutoxycarbonylation of hexene-1, at which the yields of target products reached 91.8% and 91.5%, respectively, were determined.

**Keywords:** $\alpha$-olefins; carbon monoxide; alcohols; hydroalkoxycarbonylation; palladium phosphine complexes





## 1. Introduction

The carbonylation of unsaturated substances has garnered significant attention in the realm of organic chemistry because of its importance in the production of a diverse array of valuable compounds [1,2]. The hydroalkoxycarbonylation of olefins with phosphine complexes of Group VIII metal elements from the periodic table is a versatile process for manufacturing carboxylic acid esters [3]. Carboxylic acid esters are versatile organic compounds with a wide range of applications in various industries. Their unique chemical properties make them useful in obtaining valuable compounds for a wide variety of products, in the production of flavoring and perfume compositions [4,5], as solvents [6], as plasticizers to increase the flexibility and durability of plastics [4,7] and in pharmaceuticals as intermediates for the synthesis of many medical drugs [8,9].

This process has attracted considerable attention due to its atomic economy, high selectivity and applicability to various substrates [10,11]. Palladium-based catalysts, especially in the presence of phosphine ligands, have become a powerful tool to stimulate this transformation [12]. The use of palladium phosphine complexes as catalysts in olefin hydroalkoxycarbonylation reactions has been widely investigated [10,13]. The choice of the ligand and its effect on the catalyst performance is a key point. Phosphine ligands not only stabilize the palladium center but also affect the regioselectivity of the reaction and reactivity of the reactants [14–16]. Recent studies have highlighted the development of tailored ligands to achieve superior catalytic activity, demonstrating the importance of ligand engineering in this context [15].

It is known from the literature that primary alcohols usually exhibit the highest regioselectivity in olefin hydroalkoxycarbonylation reactions [17,18]. This regioselectivity

is due to the electronic and steric factors associated with primary alcohols. Primary alcohols typically undergo Markovnikov addition to the olefin, leading to the formation of linear carbonylated products. This regioselectivity is often desirable in chemical synthesis because it allows for the preparation of valuable linear compounds [19]. Steric hindrance is lower in primary alcohols than in secondary or tertiary alcohols [20]. This makes the addition of a primary alkoxy group to an olefinic carbon less difficult and more favorable. The electronic properties of primary alcohols also play a role in their regioselectivity. The electron-donating nature of the alkyl group in primary alcohols enhances the nucleophilicity of the oxygen atom, favoring attack on the olefinic carbon [21,22].

This work is a sequential continuation of the works in which the optimal ratios of the catalytic system and reagents, as well as the parameters for carrying out the process (temperature, pressure and duration) of hexene-1 hydroethoxycarbonylation using the catalytic systems PdCl$_2$(PPh$_3$)$_2$-PPh$_3$-AlCl$_3$ [23] and Pd(PPh$_3$)$_4$-PPh$_3$-TsOH [24], were determined in detail. The objective of this study is to determine the important role of alcohols in the hydroalkoxycarbonylation reaction of hexene-1, which is crucial for the design of efficient catalytic systems and the achievement of specific synthetic compounds in organic chemistry.

## 2. Results and Discussion

### 2.1. Performance of Hexene-1 Hydroalkoxycarbonylation Reaction Using PdCl$_2$(PPh$_3$)$_2$-PPh$_3$-AlCl$_3$ Catalytic System

The catalytic activity of the catalytic system PdCl$_2$(PPh$_3$)$_2$-PPh$_3$-AlCl$_3$ was investigated in the carbonylation reaction of hexene-1 with carbon monoxide and ethanol, propanol, butanol-1, isobutanol, pentanol-1, tert-butyl alcohol, isoamyl alcohol and allyl alcohol. Each reaction resulted in the formation of two isomeric products: with ethyl alcohol, linear (ethyl ester of enanthic acid (EEEA)) and branched products (ethyl ester of 2-methylcaproic acid (EE2-MCA)); with propyl alcohol, linear (propyl ester of enanthic acid (PEEA)) and branched products (propyl ester of 2-methylcaproic acid (PE2-MCA)); with butyl alcohol, linear (butyl ester of enantoic acid (BEEA)) and branched products (butyl ester of 2-methylcaproic acid (BE2-MCA)); with isoamyl alcohol, linear (isoamyl ester of enanthic acid (IEEA)) and branched products (isoamyl ester of 2-methylcaproic acid (IE2-MCA)); with isobutyl alcohol, linear (isobutyl ester of enantoic acid (IEEA) and branched products (isobutyl ester of 2-methylcaproic acid (IE2-MCA)); and with pentyl alcohol, linear (pentyl ester of enantoic acid (PEEA)) and branched products (pentyl ester of 2-methylcaproic acid (PE2-MCA)).

The experimental results obtained are summarized in Table 1. As indicated by the information in the table, the hydroalkoxycarbonylation of hexene-1 resulted in the creation of both linear and branched products. The highest yield of the target products was observed when ethanol, propanol and butanol-1 were used as alcohols. In the case of aliphatic alcohols, the regioselectivity and yield of the target products were influenced by the radical size and structure of the starting alcohols. An increase in the radical size of the initial aliphatic alcohols decreased the total yield of the reaction products, and in the case of a branched structure of aliphatic alcohols, the total yield of the isomeric products decreased. As a result of the reaction, the following yields of products were obtained: with ethyl alcohol, 89.1%; with propyl alcohol, 91.8%; with butyl alcohol, 91.5%; with isoamyl alcohol, 46.6%; with isobutyl alcohol, 88.3%; and with pentyl alcohol, 67.6%. In reactions with allyl and tretbutyl alcohol, no products were formed.

**Table 1.** Study of the effect of catalytic system (PdCl$_2$(PPh$_3$)$_2$-PPh$_3$-AlCl$_3$) on the yield of products in the hexene-1 hydroalkoxycarbonylation reaction.

| Entry | [C$_6$H$_{12}$]:[ROH] | Product Yield, % | Selectivity iso:n |
|---|---|---|---|
| 1 | Ethanol | 89.1 | 1:2.5 |
| 2 | Propanol-1 | 91.8 | 1:2.8 |
| 3 | Butanol-1 | 91.5 | 1:2.7 |
| 4 | Isoamyl alcohol | 46.6 | 1:2.35 |
| 5 | Isobutanol | 88.3 | |
| 6 | Pentanol-1 | 67.6 | |
| 7 | Allyl alcohol | - | |
| 8 | Tert butyl alcohol | - | |

Reaction conditions: reagent molar ratio: [Hexene-1]:[ROH] = 690:435, T = 120 °C, Pco = 2.5 MPa, τ = 5 h.

## 2.2. Performance of Hexene-1 Hydroalkoxycarbonylation Reaction Using Pd(PPh$_3$)$_4$-PPh$_3$-TsOH Catalytic System

The experimental results obtained are summarized in a diagram in Table 2. The reaction of the hydroalkoxycarbonylation of hexene-1 in the case of aliphatic alcohols proceeded with the formation of linear and branched products, while with alicyclic (cyclohexanol and menthol) and benzyl alcohol, the process proceeded with 100% selectivity with respect to the linear product. The boiling points of isomeric reaction products (1) and (2) were closely situated; they could not be separated through fractional distillation. Therefore, the proportion of isomers (1) and (2) in the final products was determined using gas chromatography (GC).

**Table 2.** Study of the effect of catalytic system (Pd(PPh$_3$)$_4$-PPh$_3$-TsOH) on the yield of products in the hexene-1 hydroalkoxycarbonylation reaction.

| Entry | [C$_6$H$_{12}$]:[ROH] | Product Yield, % | Selectivity iso:n |
|---|---|---|---|
| 1 | Ethanol | 80.7 | 1:39 |
| 2 | Propanol-1 | 80.5 | 1:5.0 |
| 3 | Isopropanol | 67.8 | 1:3.6 |
| 4 | Butanol-1 | 79.0 | 1:5.3 |
| 5 | Isobutanol | 75.0 | 1:4.8 |
| 6 | Cyclohexanol | 83.1 | - |
| 7 | Mentol | 89.1 | - |
| 8 | Benzyl alcohol | 75.9 | - |

Reaction conditions: reagent molar ratio: [Hexene-1]:[ROH] = 550:435, T = 100 °C, Pco = 2.0 MPa, τ = 5 h.

The highest yields of the target products were observed for the reaction of the hydromentoxycarbonylation and hydrocyclohexoxycarbonylation of hexene-1: the yields of the menthyl and cyclohexyl esters of enanthic acid were 89.1% and 83.1%, respectively. In the case of fatty alcohols, the total yield of linear and branched products and the regioselectivity of the reaction were influenced by the size and structure of the starting alcohols. An increase in the radical size of the initial fatty alcohols decreased the total yield of the reaction products. With a branched structure of the radical of the initial aliphatic alcohol, the total yield of isomeric reaction products decreased: ethanol, 80.7%; propanol, 80.5%; iso-propanol, 67.8%; butanol, 79.0%; isobutanol, 75.0%, and benzyl alcohol, 75.9%.

Thus, it was found that the hexene-1 hydroalkoxycarbonylation reaction using Pd(PPh$_3$)$_4$-PPh$_3$-TsOH in the case of aliphatic alcohols proceeded with the formation of linear and

branched products, and in the case of alicyclic (cyclohexanol and menthol) and benzyl alcohols, regioselectivity proceeded with the formation of only linear products. The overall product yield and the reaction's regioselectivity were also impacted by the composition of the structure of fatty alcohol radicals.

*2.3. Determination of the Influence of Parameters for the Hydroalkoxycarbonylation Reaction of Hexene-1 with Propanol-1 and Butanol-1*

Earlier studies were conducted to determine the optimal parameters and the ratio of the catalytic system and reagents for the reaction of the hydroalkoxycarbonylation of hexene-1 with ethanol in the presence of the catalytic system $PdCl_2(PPh_3)_2$-$PPh_3$-$AlCl_3$ [23]. The optimal process parameters were determined for the first time for the efficient hydroalkoxycarbonylation reaction of hexene-1 with propanol-1 and butanol-1. The comparative results of the yields of (PEEA + PE2-MCA)/(BEEA + BE2-MCA) of the hydropropoxycarbonylation and hydrobutoxycarbonylation reactions of hexene-1 are summarized in Table 3.

**Table 3.** Determination of optimal parameters of the reaction process of hexene-1 hydropropoxycarbonylation and hydrobutoxycarbonylation.

| Exp. No. | $[C_6H_{12}]:[C_3H_7OH]:$<br>$[PdCl_2(PPh_3)_2]:[PPh_3]:[AlCl_3]$ | T, °C | $P_{CO}$, MPa | $\tau$, h | Products Yield,<br>(PEEA + PE2-MCA)/<br>(BEEA + BE2-MCA), % |
|---|---|---|---|---|---|
| 1 | 690:435:1:6:8 | 120 | 2.5 | 3 | 79.1/86.9 |
| 2 | 690:435:1:6:8 | 120 | 2.5 | 4 | 82.3/90.1 |
| 3 | 690:435:1:6:8 | 120 | 2.5 | 5 | 91.8/91.5 |
| 4 | 690:435:1:6:8 | 120 | 2.5 | 6 | 54.5/87.1 |
| 5 | 690:435:1:6:8 | 120 | 2.5 | 7 | 35.9/86.5 |
| 6 | 690:435:1:6:8 | 100 | 2.5 | 5 | 29.5/86.2 |
| 7 | 690:435:1:6:8 | 110 | 2.5 | 5 | 59.2/90.1 |
| 8 | 690:435:1:6:8 | 120 | 2.5 | 5 | 91.8/91.5 |
| 9 | 690:435:1:6:8 | 130 | 2.5 | 5 | 54.5/85.0 |
| 10 | 690:435:1:6:8 | 140 | 2.5 | 5 | 31.0/83.9 |
| 11 | 690:435:1:6:8 | 120 | 1.5 | 5 | 40.7/86.2 |
| 12 | 690:435:1:6:8 | 120 | 2.0 | 5 | 55.5/90.6 |
| 13 | 690:435:1:6:8 | 120 | 2.5 | 5 | 91.8/91.5 |
| 14 | 690:435:1:6:8 | 120 | 3.0 | 5 | 63.3/85.0 |

Based on the results of the obtained experimental data in Table 3, the following conclusions could be drawn.

The effect of the duration of the hydropropoxycarbonylation and hydrobutoxycarbonylation reactions of hexene-1 at a particular ratio of starting reagents and components of the catalytic system ($[C_6H_{12}]:[C_3H_7OH/C_4H_8OH]:[PdCl_2(PPh_3)_2]:[PPh_3]:[AlCl_3]$ = 690:435:1:6:8) on the yield of the target products was studied. The results of the conducted experiments are summarized in Table 3. During the experiment, under the optimum conditions of T = 120 °C and $P_{CO}$ = 2.5 MPa, the temperature was raised to 120 °C for 1 h, and the reaction mixture was kept for 3 to 7 h. It was found that, during the hydropropoxycarbonylation and hydrobutoxycarbonylation of hexene-1, the yield of products was significantly affected by changing the duration of the reaction. Increasing the process duration from 3 to 5 h increased the total yield of target products from 79.1% to 91.8% for the hydropropoxycarbonylation reaction, and for the hydrobutoxycarbonylation reaction, the total yield of target products increased from 86.9% to 91.5%, but a further increase in time up to 7 h in both cases led to a sharp decrease in the yield of formed products to 35.9% and 86.5%, respectively. With a long duration of the process, the decomposition of reaction products into initial components took place, i.e., a reversible process. Thus, on the basis of the experimental data, it was proved that the optimal time for the reaction of the hydropropoxycarbonylation and hydrobutoxycarbonylation of hexene-1 is 5 h.

The effect of temperature on the yield of the target product was studied when the reactions of the hydropropoxycarbonylation and hydrobutoxycarbonylation of hexene-1 were carried out at a particular ratio of the components of the catalytic system and reagents ($[C_6H_{12}]$:$[C_3H_7OH/C_4H_8OH]$:$[PdCl_2(PPh_3)_2]$:$[PPh_3]$:$[AlCl_3]$ = 690:435:1:6:8), with $P_{CO}$ = 2.5 MPa and $\tau$ = 5 h. The results of the conducted experiment are presented in Table 3. Increasing the process temperature from 100 °C to 120 °C increased the total yield of the target products from 29.5% to 91.8% for the hydropropoxycarbonylation reaction and from 86.2% to 91.5% for the hydrobutoxycarbonylation reaction, but in both reactions, a further increase in temperature up to 140 °C led to a sharp decrease in the yield of the formed product to 31.1% and 83.9%, respectively. At elevated temperatures, side reactions or competing pathways may become more prevalent. For example, the higher reactivity of propanol compared to butanol can lead to undesirable reactions resulting in the formation of by-products or the decomposition of key intermediates, which reduces the overall yield of the desired product. Thus, based on the experimental data, it was proved that the most optimal reaction time for the hydropropoxycarbonylation and hydrobutoxycarbonylation reactions of hexene-1 is 120 °C.

The effect of carbon monoxide pressure on the hydropropoxycarbonylation and hydrobutoxycarbonylation reactions of hexene-1 was investigated at the optimal ratio of the components of the catalytic system ($[C_6H_{12}]$:$[C_3H_7OH]$:$[PdCl_2(PPh_3)_2]$:$[PPh_3]$:$[AlCl_3]$ = 690:435:1:6:8). The results of the conducted experiment are summarized in Table 3. During the experiment, it was found that, under the optimum process conditions of T = 120 °C and $\tau$ = 5 h for the hydropropoxycarbonylation and hydrobutoxycarbonylation of hexene-1, carbon monoxide pressure from 1.5 MPa to 2.5 MPa significantly increased the total yield of target products from 40.7% to 91.8% for the hydropropoxycarbonylation reaction and from 86.2% to 91.5% for the hydrobutoxycarbonylation reaction, but a further increase in CO pressure to 3.0 MPa markedly reduced the yield of formed products in both reactions to 63.3% and 85.0%, respectively. This is apparently due to the deactivation of the catalyst by the appearance of palladium niello on the inner walls of the reactor. Based on the results of the study, it was proved that the optimal pressure of carbon monoxide for the reaction of hydropropoxycarbonylation and hydrobutoxycarbonylation of hexene-1 is 2.5 MPa.

*2.4. Mechanism of Hexene-1 Hydroalkoxycarbonylation Reaction in the Presence of $PdCl_2(PPh_3)_2$-$PPh_3$-$AlCl_3$ and $Pd(PPh_3)_4$-$PPh_3$-TsOH Catalytic System*

The hydroalkoxycarbonylation reaction of hexene-1 in the presence of the catalytic systems $PdCl_2(PPh_3)_2$-$PPh_3$-$AlCl_3$ and $Pd(PPh_3)_4$-$PPh_3$-TsOH containing $AlCl_3$ and TsOH as promoters can proceed via a "hydride" mechanism.

$AlCl_3$ and TsOH, being strong acids, can interact with alcohols to form the protons $H^+[ROAlCl_3]^-$ and $H^+[ROTsOH]^-$, as well as $[ROAlCl_3]^-$ and $[ROTsOH]^-$ weak coordination anionic complexes. The polarization of O-H bonds in alcohols under the action of strong acids makes the process possible via a hydride mechanism. The main stage of the process is considered to be the formation of a $[H\text{-}Pd]^+$ hydride complex, which allows for the subsequent catalytic cycle to take place (Figure 1). In this work, we delve into the hydroalkoxycarbonylation reaction of hexene-1 using various alcohols as reagents, catalyzed by palladium phosphine complexes, which involves the following components and steps: Hexene-1 serves as a starting material containing a carbon–carbon double bond upon which the reaction is initiated. Carbon monoxide is the key component that reacts with the palladium catalyst to form reactive compounds Pd(II)-CO [2]. Alcohols serve as a source of alkoxyl groups in the reaction [25]. These alkoxyl groups are incorporated into the final product as the oxygen atom of the carbonyl group (C=O). They also act as nucleophiles by attacking the palladium-activated carbon monoxide complex. This step leads to the formation of a palladium alkoxycarbonyl intermediate. It has been suggested that alcohols can also coordinate to the palladium center, stabilizing key intermediates in the catalytic cycle. This coordination helps prevent unwanted side reactions and promotes the desired conversion. The reaction proceeds through various steps, including oxidative

addition, migratory insertion and reductive elimination, which eventually leads to the formation of the desired product [26].

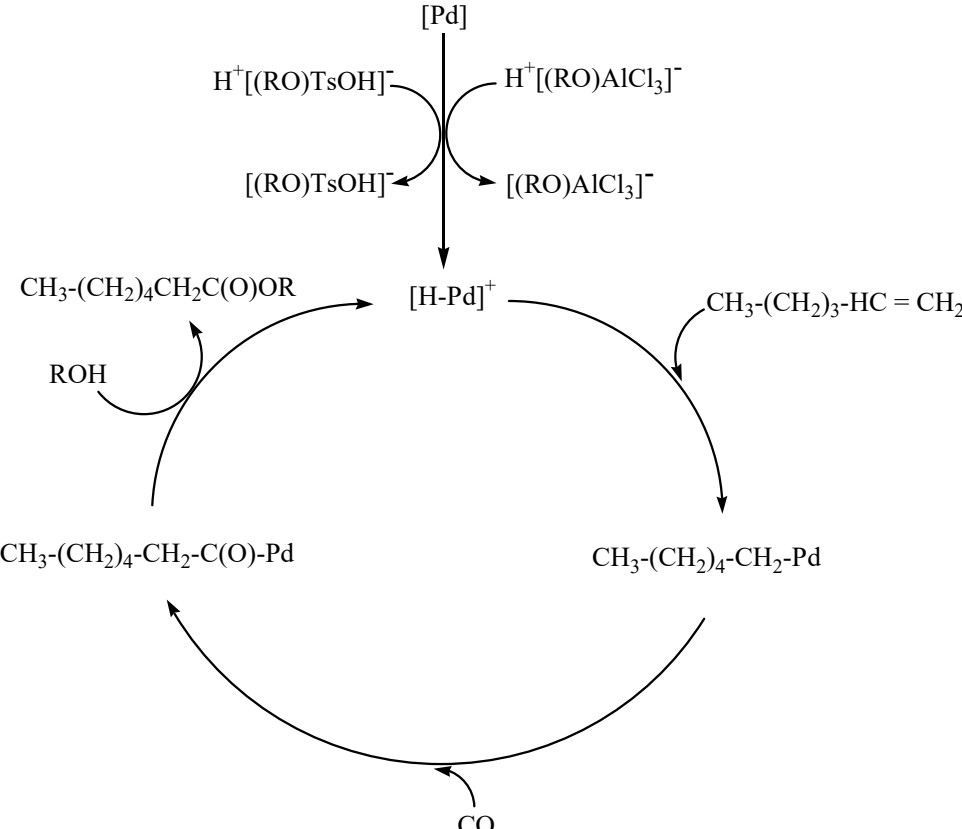

**Figure 1.** Assumed "hydride" mechanism of hexene-1 hydroalkoxycarbonylation reaction in the presence of systems based on palladium complexes with co-catalysts p-TsOH and AlCl$_3$.

## 3. Materials and Methods

### 3.1. Initial Reagents

The purchased reagents from SIGMA-ALDRICH (Saint Louis, MO, USA) (tetrakis (triphenylphosphine) palladium(0), dichlorobis(triphenylphosphine) palladium(II), p-toluene sulfonic acid, aluminum trichloride, triphenylphosphine, hexene-1) were used as starting reagents; absolutized ethanol, propanol, butanol-1, isobutanol, pentanol-1, menthol, cyclohexanol, tert-butyl alcohol, isoamyl alcohol, iso-propanol, allyl alcohol, benzyl alcohol and reagent carbon monoxide were used without special purification.

### 3.2. Methodology of the Experiment

The experiments were carried out in a 100 mL stainless steel autoclave equipped with a magnetic stirrer and a solvent-free carbon monoxide injection device (Figure 2). The starting reagents were loaded into the steel autoclave; in the example of the hydroethoxycarbonylation of hexene-1, 6.637 g (78.9 mmol) of hexene-1 and 2.289 g (49.7 mmol) of ethanol, 0.180 g (0.685 mmol) of PPh$_3$, 0.121 g (0.914 mmol) of AlCl$_3$ and 0.080 g (0.1142 mmol) of PdCl$_2$(PPh$_3$)$_2$ were used. The ratio of the starting reagents and components of the catalytic system was [C$_6$H$_{12}$]:[C$_3$H$_7$OH/C$_4$H$_8$OH]:[PdCl$_2$(PPh$_3$)$_2$]:[PPh$_3$]:[AlCl$_3$] = 690:435:1:6:8. The autoclave was sealed, purged twice with carbon monoxide to remove air and filled with 1.5 MPa carbon monoxide. The stirrer and heater were then turned on. For 1 h, the temperature was increased to 120 °C, and the carbon monoxide pressure was raised to 2.5 MPa. At a constant temperature of 120 °C and a pressure of 2.5 MPa, the reaction mixture was stirred for 5 h. Then, the stirrer and heater were turned off, and the autoclave was cooled at room temperature. Then, after cooling down, the valve was opened to release the residual pressure. After opening the reactor,

the contents of the autoclave were decanted into a container. Vacuum fractional distillation of the reaction mixture was performed to determine the yield of the target products. The total yield of products (T $_{b.p.}$ 183–200 °C) obtained via distillation was 3.85 g or 89.1%, consisting of 73.1% propyl ester of enanthic acid and 16.0% propyl ester of 2-methylcaproic acid. The products were identified using instrumental methods, such as IR, NMR and GC-MS.

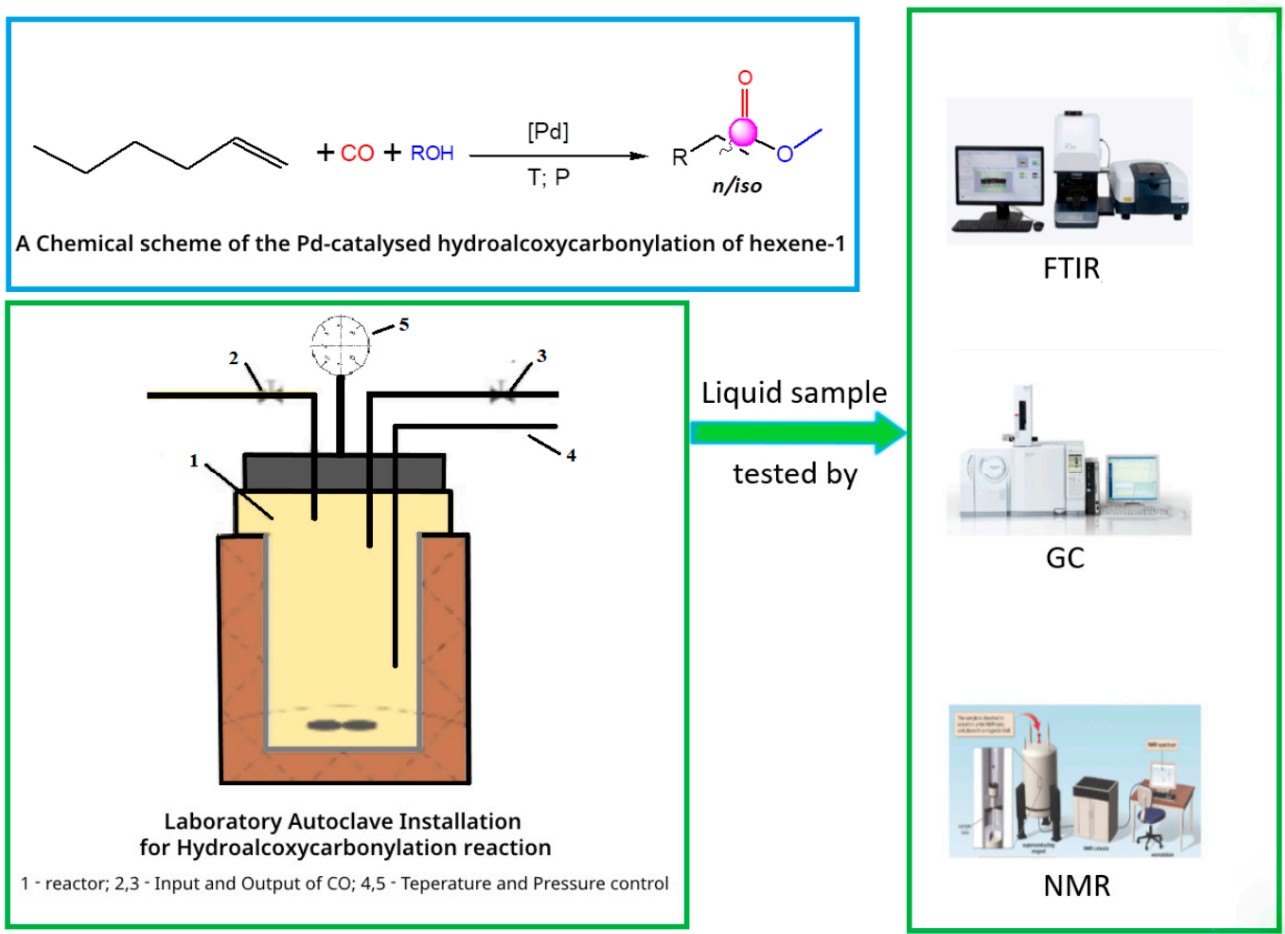

**Figure 2.** General scheme of hydroalkoxycarbonylation process of hexene-1.

### 3.3. Instrumental Analyses of the Obtained Products

The obtained isomeric linear and branched products (esters of carboxylic acids) possessed very similar boiling points, and fractional distillation could not separate them; therefore, the ratios of these two isomeric products were determined via gas chromatography on a chromatography–mass spectrometer, Agilent 7890a/5975c (Santa Clara, CA, USA). Chromatography conditions: gas chromatograph Agilent 7890a with mass-selective detector Agilent 5975c; mobile phase (gas carrier)—helium; vaporization temperature, 300 °C; flow shift (Split), 1000:1; thermostat column temperature was initially 40 °C (1 min), temperature increased 5 °C per minute until reaching 250 °C, and this temperature was applied for 1 min, with a total analysis time of 44 min; and mass detector electron impact ionization condition. HP-FFAP capillary chromatography column: column length, 30 m; inner diameter, 0.25 mm; and fixed phase, nitroterephthalic acid modified with polyethylene glycol. The identification of the synthesized compounds was carried out via IR and PMR spectroscopy. IR spectra were taken on a single-beam infrared spectrometer, "Nicolet 5700" of Termo Electron Corporation (West Palm Beach, FL, USA), in the region of 400–4000 cm$^{-1}$. $^{13}$C NMR spectra were taken on a "Varian NMR 600" (Palo Alto, CA, USA) instrument, with a working frequency of 101 MHz, using CHCl$_3$ solution as a solvent.

Under the found optimum conditions (T = 120 °C, $P_{CO}$ = 2.5 MPa and $\tau$ = 5 h) for the hydropropoxycarbonylation reaction and hydrobutoxycarbonylation reaction of hexene-1, GC analyses of the reaction mixtures were carried out (Figure 3). From the chromatograms obtained, it was observed that the reaction mixture consisted of 70.7% linear (PEEA) and 21.1% branched (PE2-MCA) products for the hydropropoxycarbonylation reaction of hexene-1, and for the hydrobutoxycarbonylation reaction of hexene-1 of 69, 8% linear (BEEA) and 22.2% branched (BE2-MCA) products, as well as unreacted residues of the starting reagents (hexene-1 and alcohols).

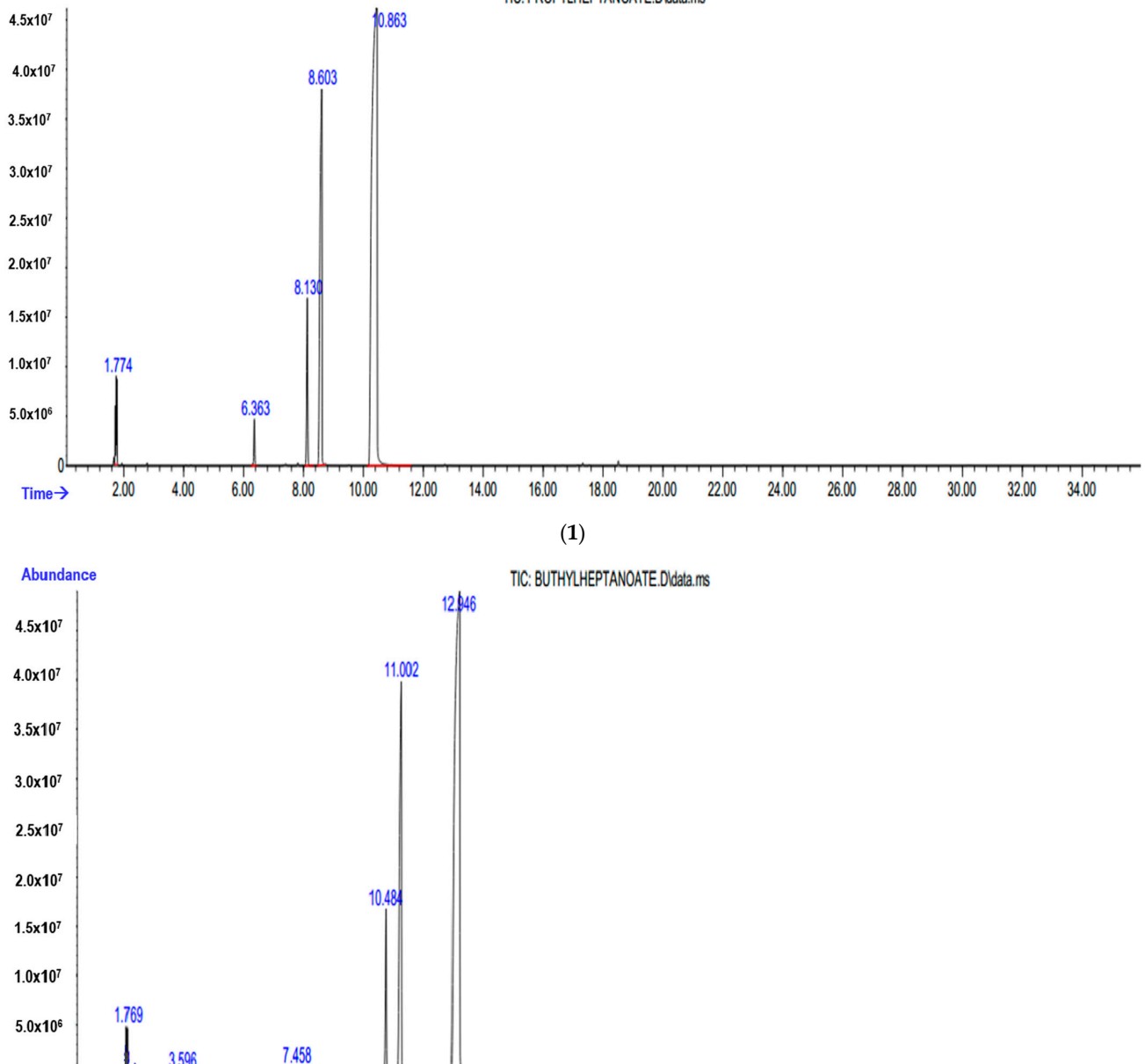

**Figure 3.** GC analysis of the reaction mixture of hydropropoxycarbonylation (**1**) and hydrobutoxycarbonylation (**2**) of hexene-1 under optimal conditions.

Figure 4 shows the IR spectra of the products of the hydropropoxycarbonylation and hydrobutoxycarbonylation reactions of hexene-1. In the IR spectrum, there is a strong absorption band at 1737 cm$^{-1}$ (C=O of the ester group); characteristic intense absorption

bands ("ether band") at 1033–1300 cm$^{-1}$; and absorption bands of CH$^{-}$, CH$_2$ and CH$_3$ groups at ~729, 1300–1462 and 2800–3000 cm$^{-1}$, respectively.

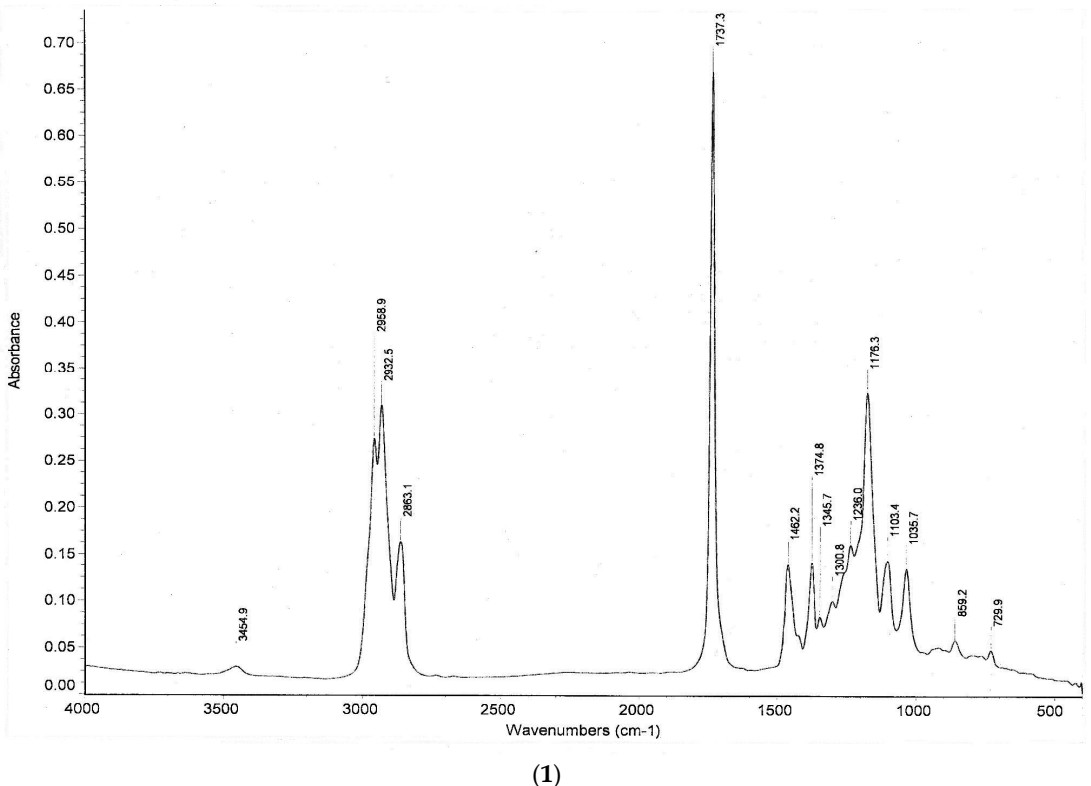

(**1**)

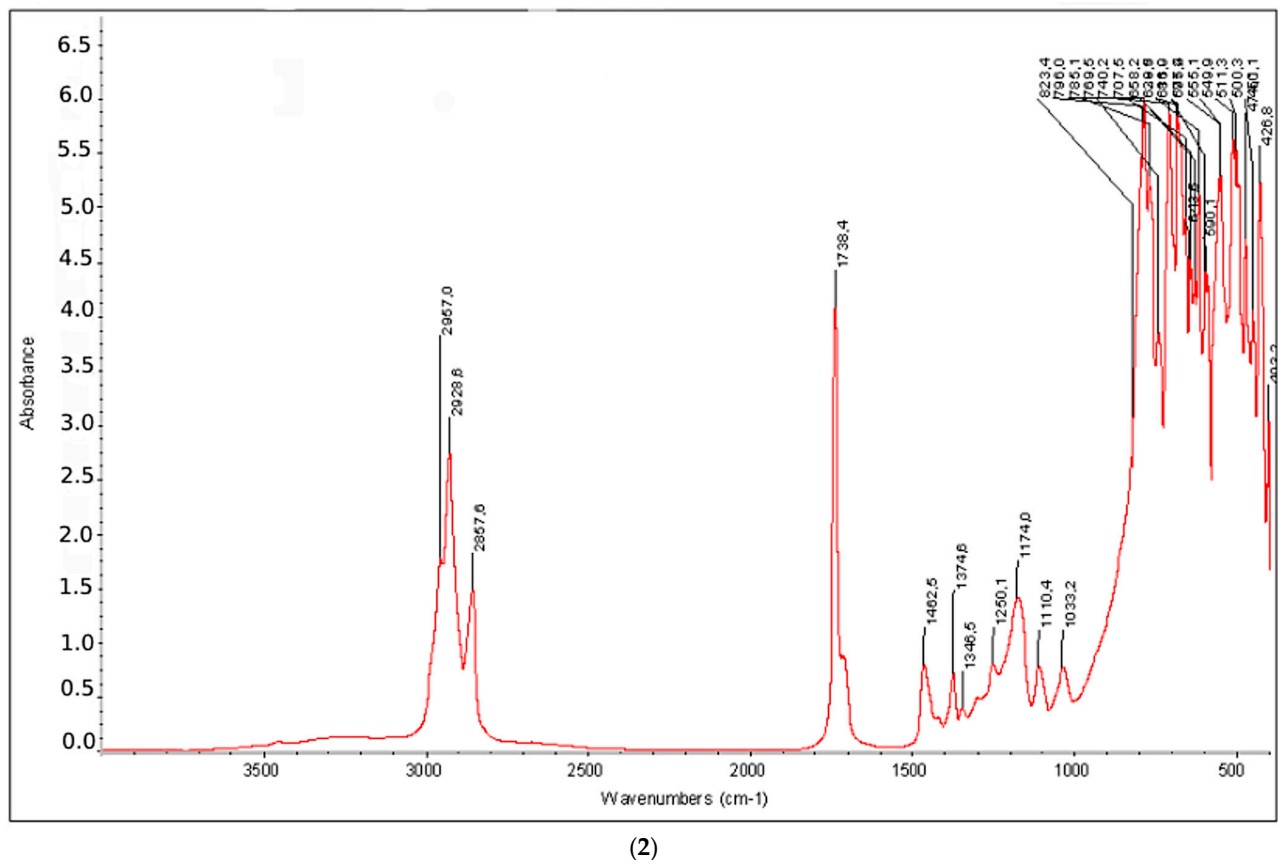

(**2**)

**Figure 4.** IR spectrum of the products obtained via the reaction of hydropropoxycarbonylation (**1**) and hydrobutoxycarbonylation (**2**) of hexene-1 under optimal conditions.

Under the optimal conditions (T = 120 °C, $P_{CO}$ = 2.5 MPa and τ = 5 h) for the reaction of the hydropropoxycarbonylation and hydrobutoxycarbonylation of hexene-1, $^{13}$C NMR spectra of the reaction mixtures were taken, and they are shown in Figures 5 and 6.

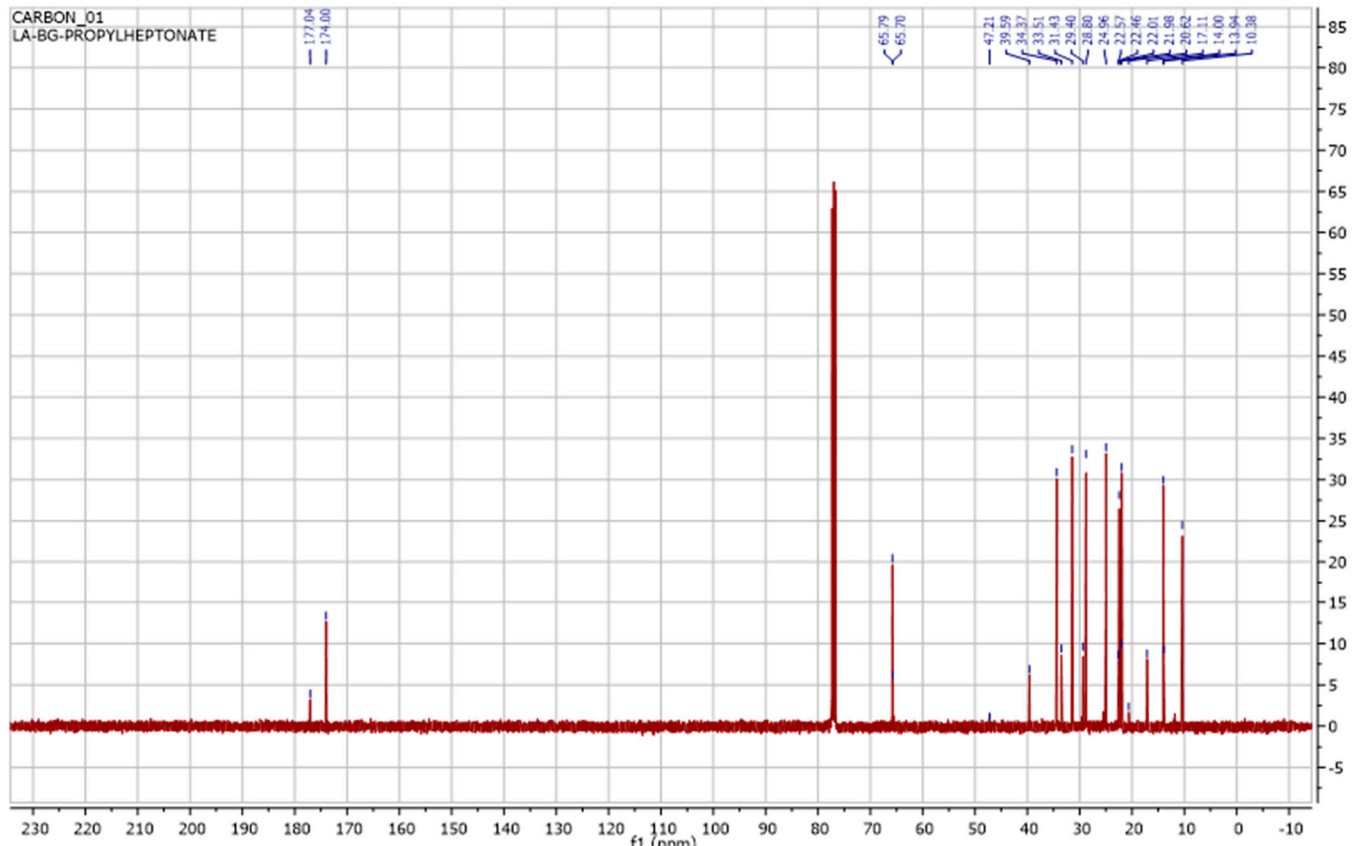

**Figure 5.** $^{13}$C NMR spectrum of the products of the hexene-1 hydropropoxycarbonylation reaction.

Figure 5 shows the $^{13}$C NMR spectrum of the product of the hexene-1 hydropropoxy-carbonylation reaction using a CHCl$_3$ solution as a solvent (at 101.53 MHz). In the $^{13}$C NMR spectrum, we observe a singlet at 174.00 ppm, which is specific to the carboxyl group (COO) carbon of the ester group, and a signal in the form of a doublet at 65.75 ppm, corresponding to 1 carbon of the propyl radical. The doublet is explained by the presence of conformational isomers. Also in the spectrum are signals at 34.47, 31.43, 28.80, 24.96, 22.57 and 22.46 ppm, typical for methylene carbons. And, finally, there are signals at 13.94 and 10.38 ppm, peculiar to terminal methyl carbons.

Thus, the $^{13}$C NMR spectrum of the reaction mixture shows a range of signals corresponding to different carbon environments within the molecule, including signals in the ester group and alkyl chain, which coincide with the hydropropoxycarbonylation reaction products of hexene-1 of linear (PEEA) and branched (PE2-MCA) structures.

Due to the fact that this work is devoted to the development of new methods for obtaining known compounds, a comparison of the IR and $^{13}$C NMR spectra with the literature data allows for the identification of compounds at a sufficient level.

Figure 6 shows the $^{13}$C NMR spectra of the products of the hexene-1 hydrobutoxycar-bonylation reaction using a CHCl$_3$ solution as a solvent (at 101.53 MHz). In the $^{13}$C NMR spectrum, we observe a singlet at 174.00 ppm, which specific to the carboxyl group (COO) carbon of the ester group, and a signal in the form of a doublet at 64.07 ppm, corresponding to 1 carbon of the butyl radical. The doublet is explained by the presence of conformational isomers. Also in the spectrum are signals at 34.38, 31.43, 29.40, 25.5, 22.57 and 22.57 ppm, typical for methylene carbons. And, finally, there are signals at 13.93 and 13.69 ppm, peculiar to terminal methyl carbons.

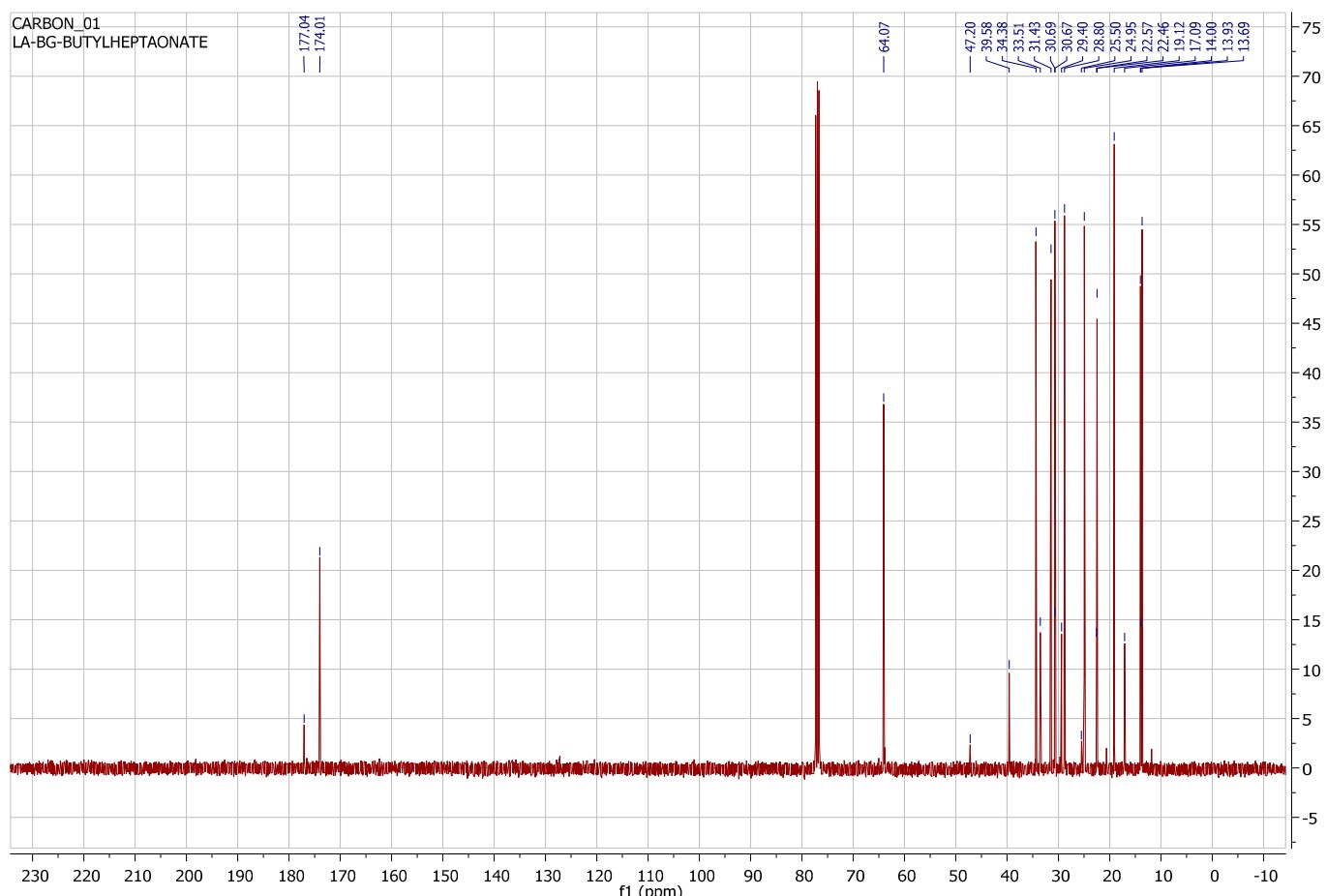

**Figure 6.** $^{13}$C NMR spectrum of products of hexene-1 hydrobutoxycarbonylation reaction.

Thus, the $^{13}$C NMR spectrum of the reaction mixture shows a range of signals corresponding to different carbon environments within the molecule, including signals in the ester group and alkyl chain, which coincide with the hydrobutoxycarbonylation reaction products of hexene-1 of linear (BEEA) and branched (BE2-MCA) structures.

## 4. Conclusions

Based on the results of the research, the following conclusions could be drawn:

1. In the carbonylation reaction of hexene-1 with carbon monoxide and alcohols (ethanol, propanol, butanol-1, isobutanol, pentanol-1 and isoamyl alcohol), the three-component catalytic system PdCl$_2$(PPh$_3$)$_2$-PPh$_3$-AlCl$_3$ exhibited high activity. As a result, under optimal conditions ([C$_6$H$_{12}$]:[ROH]:[PdCl$_2$(PPh$_3$)$_2$]:[PPh$_3$]:[AlCl$_3$] = 690:435:1:6:8, T = 120 °C, Pco = 2.5 MPa and τ = 5 h), the following yields of products were obtained: in hydroethoxycarbonylation, 89.1%; in hydropropoxycarbonylation, 91.8%; in hydrobutoxycarbonylation, 91.5%; in hydroisoalyoxycarbonylation, 46.6%; hydroisobutoxycarbonylation, 88.3%; and in hydropentoxycarbonylation, 67.6%. The reactions with allyl and tertbutyl alcohols showed no activity.

2. In the carbonylation reaction of hexene-1 with carbon monoxide and alcohols (ethanol, propanol, isopropanol, butanol-1, isobutanol, cyclohexanol, menthol and benzyl alcohol), the three-component catalytic system Pd(PPh$_3$)$_4$-PPh$_3$-TsOH showed high activity. Under the found optimal conditions ([C$_6$H$_{12}$]:[ROH]:[Pd(PPh$_3$)$_4$]:[PPh$_3$]:[TsOH] = 550:435:1:6:12, P$_{CO}$ = 2.0 MPa, T = 100 °C and τ = 5 h), the yields of the products were 80.7% in hydroethoxycarbonylation; 80.5% in hydropropoxycarbonylation; 67.8% in hydroisopropoxycarbonylation; 79.0% in hydrobutoxycarbonylation; 75.0% in hydroisobutoxycarbonyla-

tion; 83.1% in hydrocyclohexoxycarbonylation; 89.1% in hydromentoxycarbonylation; and 75.9% in hydrobenzyloxycarbonylation.

3.　The influence of the process duration, temperature and carbon monoxide pressure on the yield of target products were also investigated in detail in the hydropropoxycarbonylation and hydrobutoxycarbonylation of hexene-1 at a particular ratio of starting reagents and components of the catalytic system ($[C_6H_{12}]$:$[C_3H_7OH/C_4H_9OH]$:$[PdCl_2(PPh_3)_2]$:$[PPh_3]$:$[AlCl_3]$ = 690:435:1:6:8), and the total yields of the target products were 91.8% and 91.5%, respectively.

4.　IR, NMR and GC-MS spectra of the synthesized products were taken, and the structures were studied and analyzed.

5.　The mechanism of hexene-1 hydroalkoxycarbonylation in the presence of two catalytic systems was proposed.

**Author Contributions:** Conceptualization, G.Z. and K.S.; Data curation, N.K.; Formal analysis, M.Z.; Funding acquisition, G.A. and G.B.; Investigation, G.A.; Methodology, G.Z.; Project administration, N.K.; Resources, M.B.; Software, N.K.; Supervision, K.S.; Validation, F.K.; Visualization, G.A.; Writing—original draft, G.Z. and K.S.; Writing—review and editing, K.S. and N.K. All authors have read and agreed to the published version of the manuscript.

**Funding:** This research was funded by the Science Committee of the Ministry of Education and Science of the Republic of Kazakhstan (Grant No. AP09058656 "Development of scientific foundations for metal complex hydroalkoxycarbonylation of $C_4$–$C_{10}$ olefins in oil refining").

**Data Availability Statement:** Data are contained within the article.

**Conflicts of Interest:** The authors declare no conflict of interest.

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
