# Peer review of "The Role of Alcohols in the Hexene-1 Hydroalkoxycarbonylation Reaction with Catalysts Based on Palladium Complexes"

_catalysts, doi:10.3390/catal13121507_

Round 1

Reviewer 1 Report

Comments and Suggestions for Authors

In this manuscript, the activity of various alcohols in the hexene-1 hydroalkoxycarbonylation reaction in the presence of two catalytic systems was investigated. The optimal process parameters for the reaction of hydropropoxycarbonylation and hydrobutoxycarbonylation of hexene-1 have been determined. NMR and chromato-mass spectra of the synthesized products were taken, structure was studied and analyzed. However, the manuscript suffers from a range of issues that need to be addressed. Therefore, the manuscript could be considered for publication after a major revision.

The principal ones of which I list below:

1. The reaction mechanism is not clear. Please provide more information to demonstrate the catalysts in the role of reactions.

2. The authors need to explain the reasons for the inconsistencies in the activity of various alcohols in the hexene-1 hydroalkoxycarbonylation reaction.

3. It is necessary to determine the structure of the product by FT-IR.

4. It will be better to discuss the current challenges of palladium phosphine complexes in more detail.

5. The introduction and conclusions are too extensive, please provide a more economical presentation.

6. Figures should be arranged in the order in which it is present in the manuscript. Figures 2-5 should provide a clearer picture. Moreover, Tables 2 and 3 were a little bit confusing.

7. The authors should polish their language carefully because many mistakes can be found throughout the manuscript.

Comments on the Quality of English Language

Extensive editing of English language required

Author Response

Dear reviewer, we fully agree with your comments and thank you for your work. We have tried to take into account and correct all our shortcomings.

Reviewer 2 Report

Comments and Suggestions for Authors

The manuscript “The role of alcohols in the hexene-1 carbonylation reaction with catalysts based on palladium complexes” deals with the investigation of catalytic hydroalkoxycarbonylation of hexane-1 with different alcohols in various conditions. This is an interesting field of modern organic chemistry.

During read the manuscript, I stopped in following points:

1.      Section 2.1. In this section authors introduce several abbreviations such as EE2-MCA, PE2-MCA etc. These abbreviations are not used further. I think they should be removed from the manuscript because they make it difficult to read.

2.      Section 2.1. The authors describe the ratio of components in the catalytic system using molar units. But for reagents (hexane and alcohol) a constant volume (!) ratio is used. This is a very strange approach to reaction.

3.       Section 2.2, line 127. The authors use a new term for benzyl alcohol, “arylaliphatic” (which is also not used in the following text). I don't know why this is needed. I would like to advise the authors to familiarize themselves with the concept of Occam's razor. The relevant material is available on Wikipedia.

4.      Contrary to section 2.1 authors used for presentation of results not a table but a diagram. It complicates the understanding of material and comparison between this and previous section. If the manuscript will be accepted the diagram must be converts to table format.

5.      Signature for Fig.1. In this place of text the authors give the ratio of reagents as “[C6H12]:[ROH]:[Pd(PPh3)4]:[PPh3]:[TsOH] = 550:435:1:6:12”. I venture to guess that is a molar ratio. Is it true? Why here the authors not use constant volume ratio for reagents as in section 2.1?

6.      By the way, in lines 147 and 154 unlucky benzyl alcohol open another “new” type of organic compounds, “arylaromatic”. Please see point 3 again.

7.       Section 2.3. The phrase “During the experiment under optimal conditions T = 120 ºC, P CO = 2.5 MPa for 1 hour the temperature was raised to 120 ºC, the reaction mixture was kept for 3-7 hours.” doesn't make sense.

8.      Description of figure 2 (chromatogram contains five peaks) is absent in the text. Moreover fig. 2 not mentioned anywhere in the text.

9.      Characteristics of NMR instrument (lines 283-287) in the text are not need. They presents in corresponding chapter – 3.1.

10.  The description of NMR spectrum (lines 302-303) must be moved to experimental section.

11.   The part of text which describes the NMR spectrum of butylheptanoate is absolutely unreadable. I can’t understand what information authors want to present. If the authors want to associate the signals in spectrum I advise to register correlation spectrum like 1H-13C HMQC.

12.   In the experimental section (sections 3.2 and 3.3) not presented any constants (spectra etc.). In general, the experimental part in extremely poor.

13.   Conclusions not well characterize the obtained results, especially point 5 of its.

Thus, I do not recommend the submitted manuscript for publication. In my opinion it should be reject.

Author Response

(The authors gave the same response as above.)

Reviewer 3 Report

Comments and Suggestions for Authors

This manuscript presents quite interesting and potentially useful results on catalytic carbonylations of 1-hexene with some alcohols. Since the object of this study is in most cases referred to as “hydroalkoxycarbonylation”, this name should appear in the title instead of “carbonylation”. In the Introduction section, the goal of this study should be clearly given. Continuation of some previous works must not stand for a valid scientific goal  (“sequential continuation of the works .. “, line 75). It would be useful if the role of AlCl3/TsOH in the catalytic system was explained. It is not clear why in some experiments TsOH was used in place of AlCl3?

The Authors should make significantly more efforts to improve the scientific quality of their manuscript. In the present form, it is not suitable for publication in any scientific journal.

Major problems:

All catalytic runs should be performed at least in duplicates to ensure the reproducibility of the data. Usually, conversions are measured with +/- 2% errors, so all yields should be reported as whole numbers (89 instead of 89.06). By the way, full stop must be used as a decimal point, not comma.

Interpretation of NMR data: the 13C NMR data clearly indicate that two compounds are present in the investigated samples. Two resonances at 177.04 and 174.00 ppm most probably correspond to the iso- and n-ester, the same applies to signals in the alkyl range. The signals should be properly assigned for the two expected isomers based on their intensity.

NMR data (both 1H and 13C) should be explicitly listed in the Experimental for all compounds. The same applies to MS spectra of all products of the catalytic reactions.

GC data: all peaks should be assigned. There are five peaks on Figure 2, while the Authors discuss only two products.

Figures 3 and 5: the correct caption should be: The 13C{1H} NMR spectrum of … The list of signals should be removed from these figures and transferred to the Experimental section.

Figures 6, 7, 8: I do not see any comments/remarks in the manuscript to these graphs. Why do you think they are of interest to the scientific community? It appears that there is a sharp decrease of yield after 5h for propanol. Please explain why?

Minor problems:

Lines 297-298 “The 13C NMR spectrum of butylheptanoate shows a variety of carbon environments within the molecule, including carbonyl carbon, carbon atoms in alkyl chains, and doublets indicating bonding to protons.”

This sentence makes no sense since the spectra are proton-decoupled.

 Line 409 „chromato-mass spectra”

This is an unacceptable jargon.

 “The Varian NMR system, functioning at 600 MHz, is a high-field NMR spectrometer, offering superior resolution and sensitivity compared to lower-field apparatus. NMR samples are placed within slim-walled NMR tubes, and the sample is dissolved in a  deuterated chloroform (CDCl3) solvent to establish a reference signal while preventing  interference from solvent protons.”

Please state only the frequency and solvent, other comments are needless.

I also do not see any support/experimental evidence for the last statement in Conclusions.

Comments on the Quality of English Language

English language needs some corrections.

Author Response

(The authors gave the same response as above.)

Round 2

Reviewer 1 Report

Comments and Suggestions for Authors

This manuscript was carefully revised based on the reviewers' comments. I recommend publication of this manuscript.

Comments on the Quality of English Language

Minor editing is required for English.

Author Response

Dear reviewer, we thank you for work and your positive feedback.

Reviewer 2 Report

Comments and Suggestions for Authors

After revision and revision, the manuscript became much better. However, several questions still remain.
1. For unknown reason, the authors do not provide and described 1H NMR spectra of the resulting products anywhere.
2. Point 5 of the conclusions needs to be rewrite. For example as "The mechanism of hexene-1 hydroalkoxycarbonylation in the presence of two catalytic systems was proposed".

Author Response

Dear reviewer, we fully agree with your comments and thank you for your work. We have tried to take into account and correct all our shortcomings.

  1. For unknown reason, the authors do not provide and described 1H NMR spectra of the resulting products anywhere.

Answer to 1 question. Due to the fact that this work is devoted to the development of new methods for the preparation of esters, and not to the production of new compounds, and the resulting esters are known, we decided that the 13C NMR spectrum was sufficient for comparison with the spectra from the databases.

https://spectrabase.com/spectrum/GHKCyh0vhKQ

  1. Point 5 of the conclusions needs to be rewrite. For example as "The mechanism of hexene-1 hydroalkoxycarbonylation in the presence of two catalytic systems was proposed".

Answer to 2 question. Point 5 of the conclusions has been rewritten according to your recommendation.

Reviewer 3 Report

Comments and Suggestions for Authors

This revised manuscript indeed shows some improvement vs. the first one. However, there are some points of concern that need to be corrected.

The proposed mechanism makes very little sense to me. The way it is presented (Figure 1), it features Pd on +1 oxidation state, which is rather unusual. Probably, charge +1 is missing on the tentative hydride [H-Pd] species. This would make some sense since apparently proton is added to the neutral species [Pd] in the first step. Moreover, Pd(II) or Pd(0) complex is used as a pre-catalyst (PdCl2(PPh3)2 or Pd(PPh3)4), while only one pathway is proposed on Figure 1. The correct name for the elementary steps are: migratory insertion (not “migratory introduction”) and reductive elimination (not “reductive removal”). Overall, the mechanism of this reaction is related to the well-known hydroformylation of olefins, what should be mentioned in the text. May I suggest that the Authors first get in touch with the mechanism of hydroformylation, and then reconsider their “assumed” mechanism.

Referring to the 13C NMR data, since the presented spectra are proton decoupled, how can you explain the presence of a doublet on your spectrum? (line 355) What are the coupling nuclei?

It would be a coupling constant in Hz, not “constant binding”. Please read any textbook on NMR spectroscopy before submitting a revised manuscript.

I still do not see complete NMR and mass data for all compounds that are claimed in this report.

Figure 3: there are five peaks on the graph, while only two products are described. Please assign unambiguously the remaining three peaks.

Comments on the Quality of English Language

The language is reasonable, but some corrections would be highly appreciated for example: "we delve", line 229

Author Response

Dear reviewer, we fully agree with your comments and thank you for your work. We have tried to take into account and correct all our shortcomings.

The proposed mechanism makes very little sense to me. The way it is presented (Figure 1), it features Pd on +1 oxidation state, which is rather unusual. Probably, charge +1 is missing on the tentative hydride [H-Pd] species. This would make some sense since apparently proton is added to the neutral species [Pd] in the first step. Moreover, Pd(II) or Pd(0) complex is used as a pre-catalyst (PdCl2(PPh3)2 or Pd(PPh3)4), while only one pathway is proposed on Figure 1. The correct name for the elementary steps are: migratory insertion (not “migratory introduction”) and reductive elimination (not “reductive removal”). Overall, the mechanism of this reaction is related to the well-known hydroformylation of olefins, what should be mentioned in the text. May I suggest that the Authors first get in touch with the mechanism of hydroformylation, and then reconsider their “assumed” mechanism.

Answer. When drawing up the mechanism scheme, an error occurred in the form: Pd in the oxidation state +1, [H-Pd]+. This has been fixed. The Pd(II) or Pd(0) complex (PdCl2(PPh3)2 or Pd(PPh3)4) is used as a catalyst, to shorten the volume and not repeat the same thing, we proposed a combined mechanism. This practice is found in many articles. The names of the stages of the mechanism have been renamed based on your recommendation.

Referring to the 13C NMR data, since the presented spectra are proton decoupled, how can you explain the presence of a doublet on your spectrum? (line 355) What are the coupling nuclei?

It would be a coupling constant in Hz, not “constant binding”. Please read any textbook on NMR spectroscopy before submitting a revised manuscript.

I still do not see complete NMR and mass data for all compounds that are claimed in this report.

Answer.  Regarding the interpretation of the NMR spectra, we completely rewrote the text in consultation with an NMR spectroscopy specialist.

Figure 3: there are five peaks on the graph, while only two products are described. Please assign unambiguously the remaining three peaks.

In the description of Figure 3 at the end of the text, we indicated that the spectrum contains peaks of unreacted starting reagents.